# Transfer learning-based channel estimation in orthogonal frequency division multiplexing systems using data-nulling superimposed pilots

**Chaojin Qing** *, **Lei Dong, Li Wang, Guowei Ling, Jiafan Wang** *

School of Electrical Engineering and Electronic Information, Xihua University, Chengdu, China

* qingchj@mail.xhu.edu.cn (CQ); jifanw@gmail.com (JW)

## Abstract

Data-nulling superimposed pilot (DNSP) effectively alleviates the superimposed interference of superimposed training (ST)-based channel estimation (CE) in orthogonal frequency division multiplexing (OFDM) systems, while facing the challenges of the estimation accuracy and computational complexity. By developing the promising solutions of deep learning (DL) in the physical layer of wireless communication, we fuse the DNSP and DL to tackle these challenges in this paper. Nevertheless, due to the changes of wireless scenarios, the model mismatch of DL leads to the performance degradation of CE, and thus faces the issue of network retraining. To address this issue, a lightweight transfer learning (TL) network is further proposed for the DL-based DNSP scheme, and thus structures a TL-based CE in OFDM systems. Specifically, based on the linear receiver, the least squares estimation is first employed to extract the initial features of CE. With the extracted features, we develop a convolutional neural network (CNN) to fuse the solutions of DL-based CE and the CE of DNSP. Finally, a lightweight TL network is constructed to address the model mismatch. To this end, a novel CE network for the DNSP scheme in OFDM systems is structured, which improves its estimation accuracy and alleviates the model mismatch. The experimental results show that in all signal-to-noise-ratio (SNR) regions, the proposed method achieves lower normalized mean squared error (NMSE) than the existing DNSP schemes with minimum mean square error (MMSE)-based CE. For example, when the SNR is 0 decibel (dB), the proposed scheme achieves similar NMSE as that of the MMSE-based CE scheme at 20 dB, thereby significantly improving the estimation accuracy of CE. In addition, relative to the existing schemes, the improvement of the proposed scheme presents its robustness against the impacts of parameter variations.

## Introduction

Orthogonal frequency division multiplexing (OFDM) has been widely applied in wireless communication systems, due to its attractive solution to combat multipath fading [1]. To

**Data Availability Statement:** All relevant data are within the paper and its Supporting information files.

**Funding:** This work is supported in part by the Sichuan Science and Technology Program (Grant No. 2021JDRC0003), the Major Special Funds of Science and Technology of Sichuan Science and Technology Plan Project (Grant No. 19ZDZX0016 /2019YFG0395), the Demonstration Project of Chengdu Major Science and Technology Application (Grant No. 2020-YF09-00048-SN), the Key Scientific Research Fund of Xihua University (Grant No. Z1120941), and the Special Funds of Industry Development of Sichuan Province (Grant No. zyf-2018-056). The funders had no role in study design, data collection and analysis, decision to publish, or preparation of the manuscript.

**Competing interests:** The authors have declared that no competing interests exist.

guarantee the reliable communication in OFDM systems, channel estimation (CE) plays a critical role [2] to eliminate the impact of wireless channels, and thus inspires many CE methods, e.g., non-pilot-aided CE [3, 4] and pilot-aided CE [5]. Without employing the pilot sequence (PS), the non-pilot-aided CE saves the valuable bandwidth resources [6]. Yet the high computational complexity hinders its applications [7, 8]. With employed PS, the pilot-aided CE achieves the high estimation accuracy, and thus is favored. However, the PS in pilot-aided CE severely occupies the valuable bandwidth resources [9], degrading the spectral efficiency of OFDM systems. To avoid bandwidth resource occupation, superimposed training (ST)-based CE was proposed. In ST-based CE, the available PS is superimposed on the data sequence [10], and thus saves the bandwidth resources, facilitating its practical applications [11].

Nevertheless, the superimposition between the data sequence and PS introduces the superimposed interference, which seriously affects the performance of CE and subsequent signal detection at the receiver. To suppress the superimposed interference, the variants of ST-based CE are triggered. In these variants, data-nulling superimposed pilot (DNSP) [12] is paid much attention due to its superiority in interference avoidance. The interference between data sequence and PS is avoided by arranging the PS in the frequency bins of data-null [12]. Regretfully, some information-bearing data at certain frequency bins in DNSP are removed prior to transmission, resulting in the symbol misidentification [13]. To tackle the issue of the symbol misidentification, the classic estimation algorithms such as minimum mean square error (MMSE) and maximum likelihood are introduced into DNSP [14]. Yet they are still facing many challenges, such as the availability for the second-order statistics of the channel and noise [15], the low estimation accuracy, and the high computational complexity [16].

To improve the estimation accuracy and reduce the computational complexity, deep learning (DL)-based CE schemes have been proposed in recent years [17]. In [18], two deep neural networks (DNNs) were designed to refine the channel state information (CSI) accuracy and improve the performance of data detection, respectively. Furthermore, convolutional neural network (CNN) is also commonly used in CE [19], which improves the accuracy of CE with reduced computational complexity. However, DL-based CE for OFDM system with DNSP has not been investigated, leaving a huge blank for avoiding the occupation of bandwidth resource. More importantly, these DL-based CE schemes, e.g., [20, 21], encounter serious model mismatch. When a network model is trained at one base station but then used at another base station (new environment), the network model usually needs to be retrained [22]. In order to avoid retraining the network model, transfer learning (TL) has been developed in wireless communication systems, e.g, signal detection [23], CE [24] and CSI feed back [25, 26], etc. As one of the effective options, TL-based CE is a promising solution for the OFDM systems.

Motivated by DNSP, DL, and TL, we fuse the DL-based CE and the DNSP to improve the estimation accuracy and computational complexity, and develop a lightweight TL network to enhance network generalization. To our best knowledge, the TL-based CE for DNSP scheme in OFDM systems has not been investigated. The motivation of this paper is mainly arisen due to the following considerations:

1. Although the ST-based CE saves valuable spectrum resources, its severely superimposed interference needs to be alleviated, triggering us to employ DNSP.

2. The challenges of the estimation accuracy and computational complexity need to be tackled for the CE in DNSP, which promotes us to fuse the DNSP and DL. To this end, the estimation accuracy is improved and the computational complexity is reduced, thereby alleviating the symbol misidentification in DNSP.

3. The developed CE for DNSP scheme should possess the robustness against environmental changes and is not complex. Thus, the dilemmas of network retraining and computational complexity should be solved, impelling us to develop lightweight TL network. With the slightly increased computational complexity, the trained network has a good generalization ability.

## Related works

In this work, we investigate the TL-based CE for OFDM systems with DNSP. The related works include ST-based CE, DL-based CE, and the TL applications in wireless communication. We respectively review these works as follows.

Relative to ST scheme in [27], DNSP scheme not only improves the CE, but also improves the data detection performance [28]. However, the DNSP discards the partial information of transmitted data symbols, causing the symbol misidentification. In other words, the CE accuracy of DNSP scheme is obtained at the cost of degradation of data detection performance. Thus, the works in [13, 29–36] are proposed to tackle this issue. In [13], a partially data-dependent ST scheme was proposed to reduce symbol misidentification by researching the trade-off between interference cancellation and frequency integrity. A partial-data ST was applied to OFDM in [29], in which an interference control factor is assigned for the training sequence. An enhanced scheme of [29] was proposed in [30], which shifted the positions of PS and superimposed them onto the data symbols with the lowest power sum. In [31], authors raised a data detection approach by using an iteration approach with kernel weighted least squares (LS). Due to the increase of computational complexity, the performance of CE and data detection has been improved. [32] investigated a data coding scheme to relieve the symbol misidentification. [33] utilized a constellation rotation scheme to preserve the partial symbol information discarded by the superimposed scheme. By considering the signal sub-space technique [34], theoretically analyzed that the symbol misidentification was related to both the modulation scheme and the pilot pattern. [35, 36] respectively investigated the precoding-based approaches which can effectively retrieve the discarded symbol information. Although the symbol misidentification was investigated in [13, 29–36], the estimation accuracy and computational complexity of CE remain challenges due to the mode of PS, which has not been well tackled.

Recently, DL-based CE is proposed to improve the estimation accuracy [37]. In [22], an end-to-end approach mode was applied to OFDM system using DL. The DNN is regarded as a black box for direct signal recovery. [38] constructed a PS designer using two layer neural networks and a channel estimator using DNNs, which were jointly trained to minimize the mean square error (MSE) of CE. Based on [38], the accuracy of CE was further enhanced by using another DNN in an iterative manner [39]. In addition [40], exploited the DNN to investigate CE for doubly selective fading channels. [41] introduced a residual learning based DNN for CE, in which the computation cost is greatly reduced. Along with the use of DNN, CNN is also commonly used for CE. In [19], two CNNs were utilized to extract the coarse features and refine the CSI accuracy, respectively. [42] introduce a federated learning based framework for downlink channel-CSI prediction, which updates the global model twice by considering the local model weights and the local gradients, respectively. Inspired by CNN, a graph neural network was constructed in [43] to improve the performance of CE by extracting the correlations of the CSI. Nevertheless, these DL-based CEs still face many challenges, such as low generalization with the environment change [44], long training time, complex parameter tuning, and large memory requirements [45], etc. Relative to the general DL method, the TL features

many advantages [24, 46], e.g., huge amount of data is not required, the training time is short, and the network effectively adapts to the new environment without network retraining, etc. In [24], the TL approach was exploited to speed up new environment adaptation in low-resolution multiple-input multiple-output systems. By using direct transfer, the TL-based CE was designed in [24] to adapt the migration from one environment to another, while still encountering high computational complexity.

Although these DL-based methods [47] improve the CE accuracy and computational complexity, the DL-based CE for DNSP scheme has not been investigated and faces the challenge of network retraining. The limited TL-based CEs (e.g., [24]) show a good perspective to improve CE generalization. Yet the network architectures need to be lightweight to reduce computational complexity, and thus facilitate their practical deployments. To the best of our knowledge, the TL-based CE for DNSP scheme has not been investigated as well. To remedy the deficiencies of related works (i.e., DNSP, DL and TL) and continue their advantages, we investigate TL-based CE for DNSP scheme in OFDM systems.

### Contributions

The main contributions of this paper are summarized as follows.

1. We develop a framework for fusing DL-based CE and DNSP. To our best knowledge, the DL-based CE for DNSP scheme has not been investigated. We employ DNSP to improve the spectral efficiency, while integrate DL network to enhance estimation accuracy. Our perspective of estimation accuracy improvement is to capture the linear and nonlinear solutions, in which the initial feature for CE is first extracted by the LS-based linear estimator followed by a nonlinear DL network. With the developed framework, both the spectral efficiency and estimation accuracy are improved.

2. We introduce the lightweight TL network for DL-based CE with DNSP. From the existing investigations [24], the model mismatch (i.e., the trained network cannot adapt to the change of transmission environment) has not been well addressed by DL-based CE. With the consideration of DNSP scheme, this situation is further worsened. We introduce the lightweight TL network to tackle this issue. Although the lightweight TL network is employed, it substantiality tackles the issue of model mismatch for CE. Especially, the computational complexity and online training time are slightly increased, facilitating its practical applications.

3. We develop a novel ReCNN to improve the estimation accuracy for the CE with DNSP scheme. The developed ReCNN employs CNN to capture a solution of DL-based CE, and utilizes the lightweight TL network to enhance model generalization. Thus, not only the improvement of CE accuracy is achieved by DL-based mode, but also the issue of model mismatch is addressed by TL approach without significant increase of computational complexity. This network architecture provides a good paradigm for the transmission environment adaptability of CE.

The remainder of this paper is structured as follows: In Section II, we describe the system model. The TL-based CE using DNSP method is presented in Section III, followed by the experimental results and analysis are illustrated in Section IV. Finally, Section V concludes our work.

*Notations*: Bold face upper case and lower case letters denote matrix and vector respectively. $(\cdot)^T$, $(\cdot)^H$, denote the transpose and conjugate transpose, respectively. $\lfloor \cdot \rfloor$ represents the floor

operation. diag($\cdot$) is the diagonalization operation of matrix. $\mathbf{I}$ represents an $N \times N$ identity matrix. $\|\cdot\|_2$ is the Euclidean norm.

## System model

In this paper, we consider an OFDM system with $N$ subcarriers and $P$ pilots by using DNSP. As shown in Fig 1, the modulated signal $\mathbf{s} = [s_0, s_1, \ldots, s_{N-1}]^T$ is firstly precoded by an unitary matrix $\mathbf{W} \in \mathbb{R}^{N \times N}$, e.g., Walsh Hadamard matrix, to hold the orthogonality between the PS and the data sequence. Then, the modulated signal is nulled at equidistant positions for inserting pilots [12]. For convenience, a diagonal matrix $\mathbf{J} \in \mathbb{R}^{N \times N}$ is utilized for the nulled positions

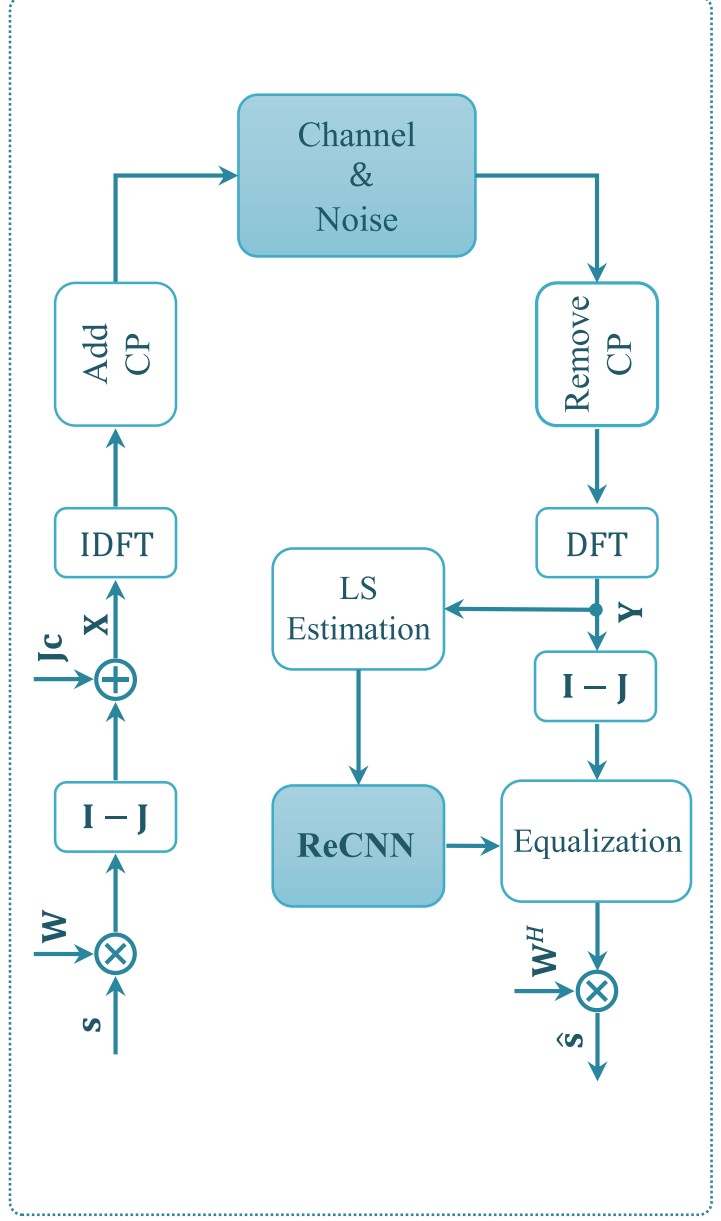

**Fig 1. System model.**

of screening, its diagonal entries are

$$[J]_{ii} = \begin{cases} 1 & i = pQ \\ 0 & i \neq pQ \end{cases}, \quad p = 0, \ldots, P - 1, \tag{1}$$

where $Q$ is the spacing between pilots with $Q = \lfloor N/P \rfloor$. Then, the transmitted signal $\mathbf{X}$ can be expressed as

$$\mathbf{X} = \sqrt{(1 - \rho)E}(\mathbf{I} - \mathbf{J})\mathbf{W}\mathbf{s} + \sqrt{\rho E}\mathbf{J}\mathbf{c}, \tag{2}$$

where $\rho \in [0, 1]$ stands for the power proportional coefficient, $E$ represents the transmitting power, and $\mathbf{c} \in \mathbb{C}^{N \times 1}$ denotes the training sequence. Then, the transmitted signal $\mathbf{X}$ is transformed to time domain by using an inverse discrete Fourier transform and added by a sufficient cyclic prefix (CP). At the receiver, after CP removal and discrete Fourier transform, the received signal $\mathbf{Y} = [y_0, y_1, \ldots, y_{N-1}]^T$ is written as

$$\mathbf{Y} = \mathbf{H}\mathbf{X} + \mathbf{n}, \tag{3}$$

where $\mathbf{H} = \mathrm{diag}(h_0, h_1, \ldots, h_{N-1})$ with diagonal entries being the frequency response of the quasi-static frequency selective fading channel, $\mathbf{n} \in \mathbb{C}^{N \times 1}$ denotes the circularly symmetric complex Gaussian (CSCG) noise with mean zero and variance $\sigma_n^2$.

With the received signal $\mathbf{Y}$, the low-complexity LS estimation is first employed to extract the initial features of CE. With the extracted features, a CNN is developed to enhance the CE, and thus structures the framework for fusing DL-based CE and DNSP. Followed by a light-weight TL network, the developed CNN further forms the ReCNN to enhance model generalization. According to the CE produced by ReCNN, we perform the equalization and detection to recovery the transmitted signal. The details of the TL-based CE are elaborated in the next section.

## Transfer learning-based channel estimation

As some information-bearing data are removed in DNSP scheme, which causes the symbol misidentification at the receiver. Meanwhile, the performance of DL-based CE is seriously degraded by the influence of varying communication environment, resulting in the need to retrain the network model. To conquer these challenges, the TL is introduced into CE by using DNSP. In the following subsections, we first introduce the TL, and then the TL-based CE is elaborated.

### Transfer learning

From [25], the same type of environment is divided into several different regions. Let $\mathcal{X}$ and $\mathcal{Y}$ denote the space of the channels in the different regions, respectively. Then, the definitions of the "domain" and the "task" are given in the following four definitions [48]:

**Definition 1**: The "domain" $\mathcal{D}$ is composed of the feature space $\mathcal{X}$ and the marginal probability distribution $P(h|h_{\mathrm{LS}})$, i.e., $\mathcal{D} = \{\mathcal{X}, P(h|h_{\mathrm{LS}})\}$, where $h$ is the real CSI and $h_{\mathrm{LS}}$ is the estimated CSI by using the LS algorithm, respectively. Meanwhile, the "task" $\mathcal{T}$ is defined as the prediction of the target channels from the source channels. Given the specific domain $\mathcal{D}$, the "task" $\mathcal{T}$ is composed of the label space $\mathcal{Y}$ and the prediction function $\mathcal{F}$, i.e., $\mathcal{T} = \{\mathcal{Y}, \mathcal{F}\}$. The prediction function $\mathcal{F}$ can be learned from the training data of the source region and then be used to predict the target channels in the target region.

Classical TL consists of two aspects, namely, the source domain transfer and the target domain adaption. Based on [48], the definition of TL can be given as follows:

**Definition 2**: Given the source task $\mathcal{T}_S$, the source domain $\mathcal{D}_S$, the target task $\mathcal{T}_T$, and the target domain $\mathcal{D}_T$, the aim of TL is to improve the performance of the target task $\mathcal{T}_T$ by using the knowledge from $\mathcal{T}_S$ and $\mathcal{D}_S$, where $\mathcal{D}_T \neq \mathcal{D}_S$ or $\mathcal{T}_T \neq \mathcal{T}_S$.

Here we extend the single-source domain transfer to the multi-source domain transfer. Then, a more generalized definition of TL can be provided as follows:

**Definition 3**: Let $\mathbb{B}_T$ and $\mathbb{B}_k$ represent respectively the set in the target and the $k$-th source given environments, where $k = 1, \ldots, K_s$, with $K_s$ being the number of source tasks. Given the source tasks $\{\mathcal{T}_S(k)\}_{k=1}^{K_s}$, the source domains $\{\mathcal{D}_S(k)\}_{k=1}^{K_s}$, the target task $\mathcal{T}_T$, and the target domain $\mathcal{D}_T$, the aim of TL is to improve the performance of the target task $\mathcal{T}_T$ by using the knowledge from $\{\mathcal{T}_S(k)\}_{k=1}^{K_s}$ and $\{\mathcal{D}_S(k)\}_{k=1}^{K_s}$, where $\mathcal{D}_T \neq \mathcal{D}_S(k)$ or $\mathcal{T}_T \neq \mathcal{T}_S(k)$. In *Definition 3*, the condition $\mathcal{D}_T \neq \mathcal{D}_S(k)$ means that either the corresponding feature space $\mathcal{X}_T \neq \mathcal{X}_S(k)$ holds or the corresponding marginal probability distribution $P_T(h|h_{LS})|_{h \in \mathbb{B}_T} \neq P_{S(k)}(h|h_{LS})|_{h \in \mathbb{B}_k}$ holds. The condition $\mathcal{T}_T \neq \mathcal{T}_S(k)$ means that either the label space $\mathcal{Y}_T \neq \mathcal{Y}_S(k)$ holds or the corresponding conditional probability distribution $P_T(h|h_{LS})|_{h \in \mathbb{B}_T} \neq P_{S(k)}(h|h_{LS})|_{h \in \mathbb{B}_k}$ holds. Since the conditional probability distributions for different prediction tasks are different, the condition $\mathcal{T}_T \neq \mathcal{T}_S(k)$ is satisfied. Therefore, the target region channel prediction for the OFDM systems can be formulated as a TL problem.

In this paper, TL transfers the knowledge using the ReCNN, which is defined as follows:

**Definition 4**: Given a TL task described by $\langle \{\mathcal{T}_S(k)\}_{k=1}^{K_s}, \{\mathcal{D}_S(k)\}_{k=1}^{K_s}, \mathcal{T}_T, \mathcal{D}_T \rangle$, it is a TL task when the prediction function $\mathcal{F}_T$ of $\mathcal{T}_T$ is a non-linear function that is approximated by the ReCNN network.

Based on *Definition 3* and *Definition 4*, the target region channel prediction for OFDM systems can be formulated as a typical TL problem, where the $k$-th learning task is to predict the *target channel* from the *source channel* in the $k$-th region.

## TL-based CE

To enhance the estimation accuracy and model generalization for the CE of DNSP scheme, a CNN is first developed followed by a lightweight TL network, and thus the ReCNN is structured. Although the ReCNN includes the DL and TL networks, the ReCNN-based CE is referred to as TL-based CE in this paper to highlight its transfer characteristics. By using the TL-based CE, not only the requirement of second-order statistics about the channel and noise is avoided, but also the adaptability of CE against varying communication environments is improved. In this sub-section, the model structure of ReCNN is briefly described, and then model training is illustrated in detail.

**Model structure.** In this paper, we fuse the LS estimator and DL-based CE for the DNSP scheme to capture the NN and non-NN solutions, respectively. To further address the model mismatch of DL-based CE, a lightweight TL network which is inspired by the DnCNN in [19] and named as ReCNN, is proposed in this paper. The modified structure is shown in Fig 2. Therein, the ReCNN consists of fourteen convolutional layers and two fully connected layers. The convolutional layers are used for extracting features from input. Then, the fully connected layers learn the non-linear combinations of these extracted features to further improve the

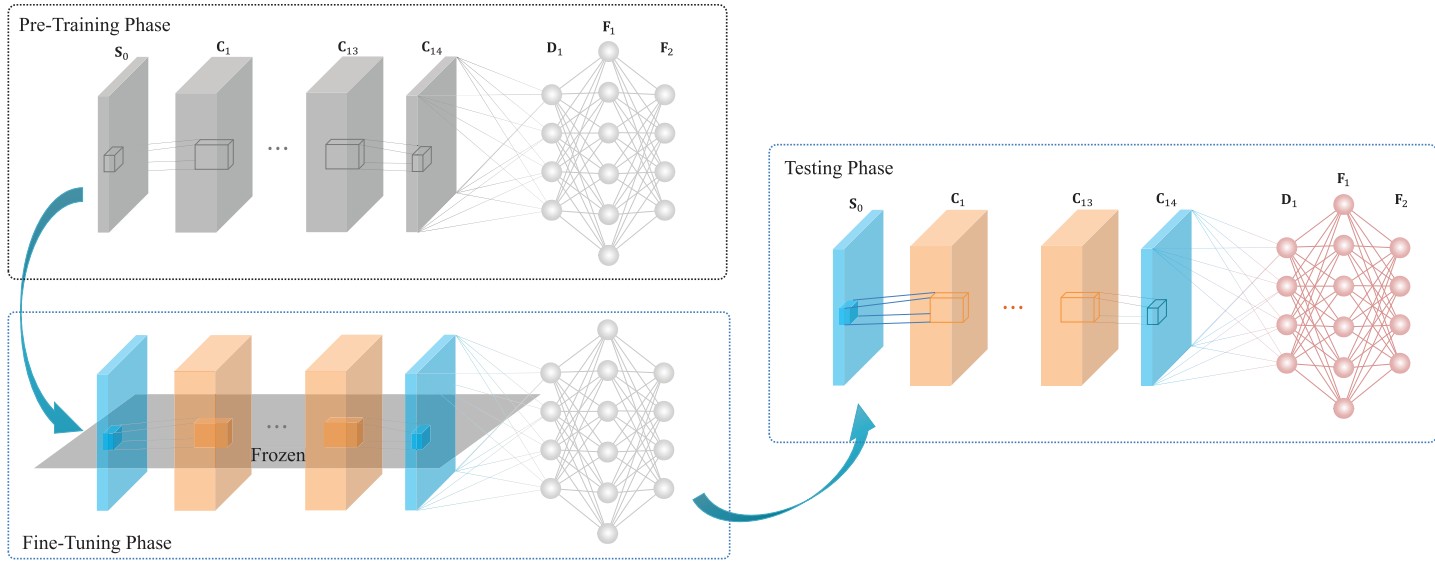

**Fig 2. The proposed ReCNN-based TL scheme for model training.**

performance of the task. In addition, the residual block between the input layer and the last convolutional layer utilizes the CNN with a subtraction structure to learn the residual noise from the noisy channel matrix for denoising. The corresponding hyperparameters are introduced in Table 1, where we first set these hyperparameters via the ReCNN structure and then fine-tune these hyperparameters to search the appropriate values for the network empirically.

**Model training.** Before introducing the ReCNN network training, LS-based CE is first described and then followed by the data collection. At last, the proposed ReCNN-based TL training, i.e., pre-training and fine-tuning are explained. The model training details are summarized in Fig 2.

- **LS channel estimation**: LS-based CE is utilized to extract an initial feature of CE in the frequency domain [49], which also serves as the data sets (including the training set, validation set and testing set) for the ReCNN. According to the LS algorithm [12], the frequency

**Table 1. Hyperparameters of the proposed ReCNN.**

**Input**: The Channel Estimated using LS Algorithm (Dimension: $N \times M \times 1$)

| Layer | (Kernel Size)×Filters | Dimension 1.5 |
|---|---|---|
| $S_0$ | – | $N \times M \times 1$ |
| $C_1$ + ReLU | $48 \times (5 \times 5)$ | $N \times M \times 48$ |
| $C_2$ + ReLU + BN | $48 \times (5 \times 5)$ | $N \times M \times 48$ |
| ⋮ | ⋮ | ⋮ |
| $C_{13}$ + ReLU + BN | $48 \times (5 \times 5)$ | $N \times M \times 48$ |
| $C_{14}$ + Linear | $1 \times (5 \times 5)$ | $N \times M \times 1$ |
| $D_1$ | – | $MN \times 1$ |
| $F_1$ + Linear | $5120 \times MN$ | $5120 \times 1$ |
| $F_2$ + Linear | $MN \times 5120$ | $MN \times 1$ |

**Output**: Prediction Channel (Dimension: $MN \times 1$)

domain channel estimator $\hat{\mathbf{H}}_P \in \mathbb{C}^{P \times 1}$ is given as

$$\hat{\mathbf{H}}_P = \left[\frac{Y(0)}{c(0)}, \frac{Y(Q)}{c(Q)}, \ldots, \frac{Y((P-1)Q)}{c((P-1)Q)}\right]^T, \tag{4}$$

where $Y(i)$ and $c(i)$, $i = 0, \ldots, N-1$, are the frequency domain values of the received signal $\mathbf{Y}$ and the PS $\mathbf{c}$, respectively. Then, we transform $\hat{\mathbf{H}}_P$ into time domain, and obtain the CE $\hat{\mathbf{h}} \in \mathbb{C}^{P \times 1}$ in time domain, i.e.,

$$\hat{\mathbf{h}} = \mathbf{F}_P^H \hat{\mathbf{H}}_P. \tag{5}$$

By adding zero at the end of $\hat{\mathbf{h}}$, a vector $\tilde{\mathbf{h}}$ with length $N$ is formed, i.e.,

$$\tilde{\mathbf{h}} = \left[\hat{\mathbf{h}}^T, \underbrace{0, \ldots, 0}_{N-P}\right]^T. \tag{6}$$

Then, the LS estimation $\hat{\mathbf{H}}_{\mathrm{LS}}$ is obtained by transforming $\tilde{\mathbf{h}}$ into the frequency domain, i.e.,

$$\hat{\mathbf{H}}_{\mathrm{LS}} = \mathbf{F}_N \tilde{\mathbf{h}}. \tag{7}$$

- **Data collection**: For the training of the ReCNN, we continuously collect $M$ time slots as one training sample, then its training set is defined as $\{\mathbb{D}\} = \{\tilde{\mathbf{H}}_{\mathrm{LS}}, \tilde{\mathbf{H}}_{\mathrm{Label}}\}$. Considering the quasi-static frequency selective fading channel [12], $\mathbf{h}$ is generated according to the widely adopted channel model COST2100 [50] without loss of generality, where the different out-door semi-urban scenarios at the 300MHz band are considered. In other words, the number of clusters in each environment are randomly distributed rather than fixed. Then, $\mathbf{h}$ is transformed into the frequency domain to form label $\mathbf{H}_{\mathrm{Label}}$, and we map the complex-valued $\mathbf{H}_{\mathrm{Label}}$ to real-valued $\tilde{\mathbf{H}}_{\mathrm{Label}}$. After we obtain the received signal $\mathbf{Y}$ from (3), $\hat{\mathbf{H}}_{\mathrm{LS}}$ is generated according to (4)–(7). Finally, the complex-valued set of $\{\hat{\mathbf{H}}_{\mathrm{LS}}\}$ is reshaped to the real-valued set $\{\tilde{\mathbf{H}}_{\mathrm{LS}}\}$.
  Considering TL-based CE in this paper, we use the training set $\{\mathbb{D}_{\mathrm{S}}(k)\}_{k=1}^{K_s}$ to represent the source domain dataset, and the $\{\mathbb{D}_{\mathrm{T}}\}$ to represent the target domain dataset. It is worth noting that the two datasets are both generated in source regions and target region of the COST2100 channel model environment, respectively. In addition, to validate the trained network parameters during the training phase, a validation set is also generated by using the same generation method of training set, and thus we can capture a set of optimized network parameters.

- **Pre-training**: We denote the whole network parameters as $\Theta = \{\Theta_{\mathrm{pre}}, \Theta_{\mathrm{fin}}\}$ of which the $\Theta_{\mathrm{pre}}$ and $\Theta_{\mathrm{fin}}$ are the sets of parameter values for *Pre-training* and *Fining-training* phase, respectively. From *Algorithm 1*, the source domain dataset $\{\mathbb{D}_{\mathrm{S}}\}$ generated by $K_s$ source tasks is utilized to train the ReCNN. To extract the time correlation, we have collected the data of $M$ time slots continuously. That is, the input of the ReCNN is the CSI values matrix $\tilde{\mathbf{H}}_{\mathrm{LS}}$ and output is the estimated channel matrix, which is denoted as

$$\hat{\mathbf{H}} = \mathcal{F}(\Theta_{\mathrm{pre}}; \tilde{\mathbf{H}}_{\mathrm{LS}}), \tag{8}$$

where $\mathcal{F}$ is the pre-training function and $\Theta_{\mathrm{pre}}$ is the pre-training parameter to be updated by training.

The total loss function of the network is the MSE between the estimated and the actual channel responses calculated as follows:

$$L_{\text{pre}} = \frac{1}{T} \| \hat{\mathbf{H}} - \tilde{\mathbf{H}}_{\text{Label}} \|_2^2, \tag{9}$$

where $T$ denotes the number of training samples. After the $\Theta_{\text{pre}}$ is obtained via $G_{\text{pre}}$ steps of ADAM updating, we use the target domain dataset $\{\mathbb{D}_{\text{T}}\}$ to test. Meanwhile, the corresponding normalized mean square error (NMSE) is saved accordingly.

- **Fine-tuning**: After the pre-training is performed according to *Algorithm 1*, we need to build the training set of the target domain. First, we divide the dataset $\{\mathbb{D}_{\text{T}}\}$ into $\{\mathbb{D}_{\text{TTr}}\}$ and $\{\mathbb{D}_{\text{TTe}}\}$. Then, the pre-trained network parameters $\Theta_{\text{pre}}$ are loaded into the ReCNN. It is worth noting that we need to freeze the parameters of convolutional layers and only update the parameters of the fully connected layers by using the backpropagation algorithm. In addition, compared with DL-based network training, much fewer samples and shorter training time are needed in the fine-tuning phase. Similar to the pre-training phase, we utilize the same loss function (given in (9)) to operate the fine-tuning phase. After the fine-tuning is finished by $G_{\text{fin}}$ steps updating, the ReCNN parameters $\Theta_{\text{fin}}$ are optimized. Finally, the testing set $\{\mathbb{D}_{\text{TTe}}\}$ is used to predict the CSI of the target environment.

**Algorithm 1**: Transfer learning for channel estimation

**Input:** The source tasks $\{\mathcal{T}_{\text{S}}(k)\}_{k=1}^{K_s}$, target tasks $\mathcal{T}_{\text{T}}$, pre-training learning rate $\gamma_1$, fine-tuning learning rate $\gamma_2$, batch size $V$, number of gradsteps for pre-training $G_{\text{pre}}$, and number of gradsteps for fine-tuning $G_{\text{fin}}$

**Output:** The pre-trained network parameter $\Theta_{\text{pre}}$, and the estimated CSI based on TL $\hat{\mathbf{H}}$.

**1 Pre-training stage**
**2** Randomly initialize the network parameters $\Theta_{\text{pre}}$
**3** Generate the training dataset $\{\mathbb{D}_{\text{S}}\} \in \{\mathcal{D}_{\text{S}}(k)\}_{k=1}^{K_s}$ for $K_s$ source tasks
**4 for** $t = 1, \ldots, G_{\text{pre}}$ **do**
**5**   Randomly select $V$ training samples from $\{\mathbb{D}_{\text{S}}\}$ as the training batch $\{\mathbb{D}_{\text{STrB}}\}$
**6**   Update $\Theta_{\text{pre}}$ by using the ADAM algorithm (learning rate $\gamma_1$) to minimize $L_{\text{pre}}$
**7 end**
**8 No-transfer testing stage**
**9** Load the trained parameters $\Theta_{\text{pre}}$ and generate the testing dataset $\{\mathbb{D}_{\text{T}}\}$
**10** Predict the target channel in the target given environment base on $\Theta_{\text{pre}}$ and $\{\mathbb{D}_{\text{T}}\}$ using (8)
**11 Fune-tuning stage**
**12** Load the pre-trained network parameters $\Theta_{\text{pre}}$
**13** Generate the fine-tuning dataset $\{\mathbb{D}_{\text{T}}\}$, and then divide $\{\mathbb{D}_{\text{T}}\}$ into $\{\mathbb{D}_{\text{TTr}}\}$ and $\{\mathbb{D}_{\text{TTe}}\}$
**14 for** $t = 1, \ldots, G_{\text{fin}}$ **do**
**15**   Load the network parameters $\Theta_{\text{pre}} \rightarrow \Theta_{\text{fin}}$
**16**   Randomly select $V$ training samples from $\{\mathbb{D}_{\text{TTr}}\}$ as the training batch $\{\mathbb{D}_{\text{TTrB}}\}$
**17**   Update $\Theta_{\text{fin}}$ by using the ADAM algorithm (learning rate $\gamma_2$) to minimize $L_{\text{fin}}$
**18 end**
**19 Transfer testing stage**
**20** Predict the CSI of target given environment base on $\{\mathbb{D}_{\text{TTe}}\}$ and parameters $\Theta_{\text{fin}}$.

## Experimental analysis

In this section, numerical results of the proposed TL-based CE using DNSP are given. First, basic parameters and definitions involved in the simulations are given. Then, the CE's NMSE and the detection's bit error rate (BER) of the proposed scheme are shown to verify the effectiveness of the proposed TL-based CE. Finally, we discuss the robustness of the proposed scheme against the influence of different parameters. The source code is available at https://github.com/Leiunnn/TransferLearningBasedCEbyDNSP.git.

### Parameter setting

In the experiments, the following basic parameters are applied unless otherwise specified. $N = 256$, $P = 8$ [12]. The channel $\mathbf{h}$ is generated by channel model COST2100 [50] at the 300MHz band, the number of multi-path is set as $L = 8$, and we collect the data of $M = 16$ time slots continuously [19]. The transmitted data symbol $\mathbf{s}$ is modulated by QPSK modulation. The data sets $\{\mathbb{D}_S\}$ and $\{\mathbb{D}_T\}$ have 8,000 and 500 samples, respectively. For both of them, the batch sizes are set as 20 samples. In pre-training phase, the data set $\{\mathbb{D}_S\}$ is divided into training set and validation set with the sizes of 6000 and 2000, respectively. Only 500 samples are employed for the $\{\mathbb{D}_T\}$, in which 300 and 200 samples are respectively allocated to $\{\mathbb{D}_{TTr}\}$ and $\{\mathbb{D}_{TTe}\}$. We use Adam optimizer as the training optimization algorithm [51] with parameters $\beta_1 = 0.99$ and $\beta_2 = 0.999$ [52]. Both the learning rates of the two phases are all set to 0.0001, and the signal-to-noise-ratio (SNR) in decibel (dB) is defined as

$$\text{SNR} = 10 \log_{10}\left(\frac{E}{\sigma_v^2}\right), \tag{10}$$

where $E$ is the transmitted power of $\mathbf{X}$, which is equal to the summation of data-symbol power $E_s$ and training-sequence power $E_c$. In these simulations, $E_s = (1 - \rho)E$ and $E_c = \rho E$, where $\rho = 0.2$. For network training, the mixed SNR is adopted, i.e., each training sample is generated under a random SNR from SNR = 0dB to SNR = 35dB with the interval of 5dB.

The NMSE is utilized to evaluate the CE performance, and defined as [53]

$$\text{NMSE} = \frac{\|\hat{\mathbf{H}} - \mathbf{H}\|_2^2}{\|\mathbf{H}\|_2^2}. \tag{11}$$

For the convenience of expression, the simplified expressions in the simulations are given as follows.

- "Proposed" is used to denote the proposed TL-based CE, "No_Transfer" denotes the proposed scheme without transferring. "LS_CE [12]" and "MMSE_CE [12]" represent "LS channel estimation in [12]" and "MMSE channel estimation in [12]", respectively. Meanwhile, "LS_CE [29]" and "MMSE_CE [29]" represent "LS channel estimation in [29]" and "MMSE channel estimation in [29]", respectively.

- "LS_CE+ZF_SD [12]", "LS_CE+MMSE_SD [12]", "MMSE_CE+ZF_SD [12]", and "MMSE_CE+MMSE_SD [12]" stand for the "LS channel estimation followed by zero forcing (ZF) equalization in [12]", "LS channel estimation followed by MMSE equalization in [12]", "MMSE channel estimation followed by ZF equalization in [12]", and "MMSE channel estimation followed by MMSE equalization in [12]", respectively. Meanwhile, "LS_CE+ZF_SD [29]", "LS_CE+MMSE_SD [29]", "MMSE_CE+ZF_SD [29]", and "MMSE_CE+MMSE_SD [29]" stand for the "LS channel estimation followed by ZF equalization in [29]", "LS channel estimation followed by MMSE equalization in [29]", "MMSE channel estimation followed by

ZF equalization in [29]", and "MMSE channel estimation followed by MMSE equalization in [29]", respectively.

- "Proposed+ZF_SD", "Proposed+MMSE_SD", "No_Transfer+ZF_SD", and "No_Transfer +MMSE_SD" stand for the "proposed transfer learning channel estimation followed by ZF equalization", "proposed transfer learning channel estimation followed by MMSE equalization", "without transfer learning channel estimation followed by ZF equalization", and "without transfer learning channel estimation followed by MMSE equalization", respectively.

## CE and Symbol Detection (SD) performance

To validate the effectiveness of proposed TL-based CE using DNSP, the performances of CE and SD under different SNRs are illustrated in Figs 3 and 4, respectively. The NMSE curves of different CE methods are compared in Fig 3, and partial numerical results are presented in Table 2 for the convenience of comparison. From Fig 3, the NMSE of [29] is higher than that of [12]. The reason is the partially data-dependent ST scheme is employed in [29], which superimposes the data sequence on the transmitted PS and thus introduces the superimposed interference into its CE. It could be observed that the NMSE of "No_Transfer" is smaller than that of "LS_CE [12]" and "LS_CE [29]". Moreover, from Table 2, the NMSE of "No_Transfer" is smaller than that of "MMSE_CE [12]" in the relatively low SNR region (e.g., SNR≤10dB).

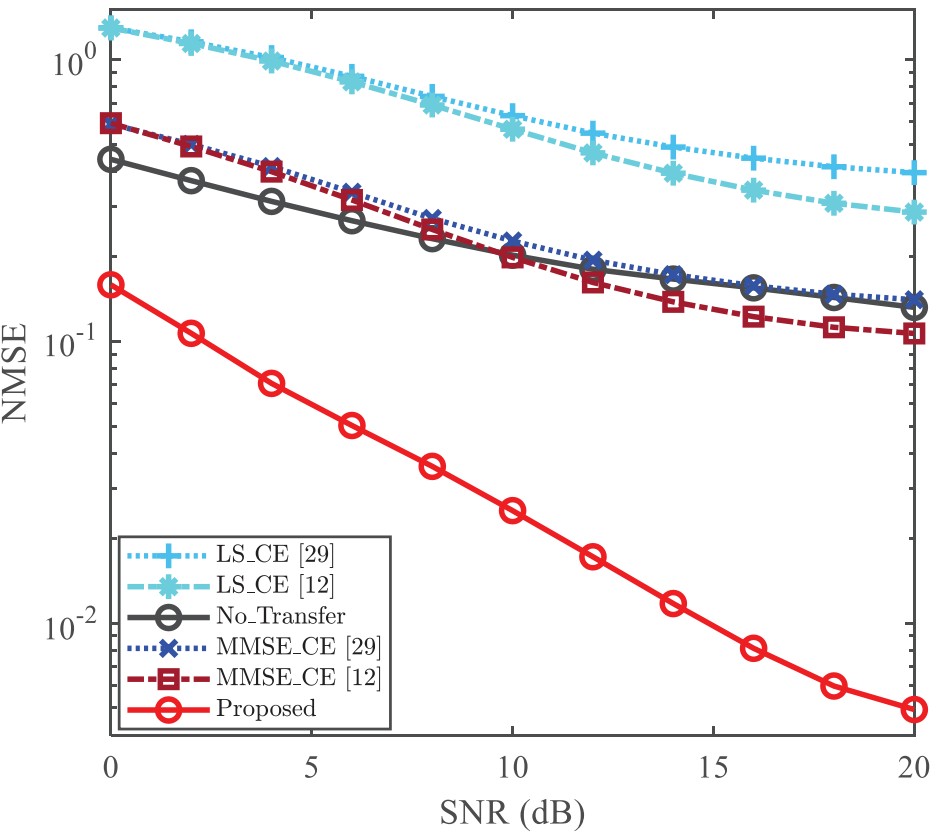

**Fig 3. NMSE vs. SNR, where $N$ = 256, $L$ = 8, $P$ = 8, and $\rho$ = 0.2.**

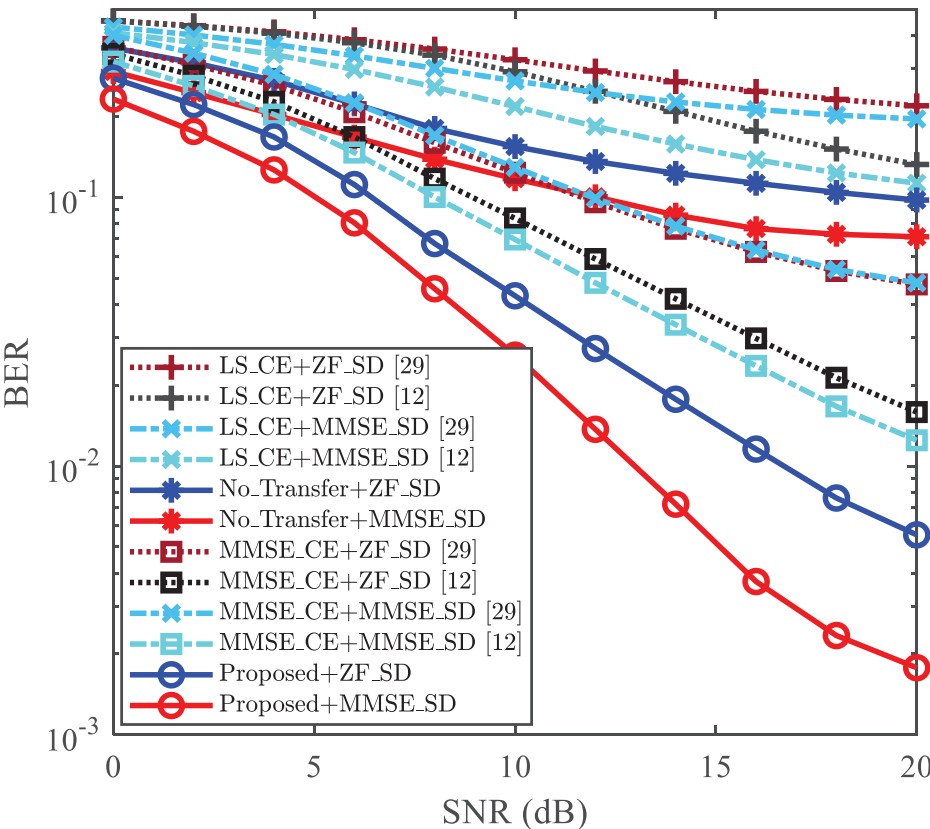

**Fig 4. BER vs. SNR, where *N* = 256, *L* = 8, *P* = 8, and *ρ* = 0.2.**

This embodies that ReCNN can still extract certain CSI features in the case where the environment is changed. From Fig 3 and Table 2, the NMSE of "Proposed" reaches the minimum in all SNR region, even compared with the "MMSE_CE [12]". For example, the NMSE of "Proposed" is $4.9 \times 10^{-3}$ for the case of SNR = 20dB, while the NMSE of "MMSE_CE [12]" is $1 \times 10^{-1}$. This reflects that the "Proposed" obtains higher CE accuracy than "MMSE_CE [12]", and thus it can work well in the varied environment. Thus the Proposed scheme possesses its effectiveness to improve the NMSE of CE.

Since the PS **c** is superimposed on the modulated data symbol **s**, it needs to be verified whether the superimposed interference (from the ST) degrades the detection performance of data symbols. In this paper, the BER is used to measure the detection performance and is

**Table 2. The NMSE of each method.**

| SNR | 0dB | 2dB | 4dB | 6dB | 8dB | 10dB | 12dB | 14dB | 16dB | 18dB | 20dB |
| --- | --- | --- | --- | --- | --- | --- | --- | --- | --- | --- | --- |
| **Method** | | | | | | | | | | | |
| LS_CE [29] | 1.295 | 1.154 | 1.015 | 0.871 | 0.739 | 0.632 | 0.547 | 0.488 | 0.446 | 0.416 | 0.396 |
| LS_CE [12] | 1.292 | 1.138 | 0.986 | 0.832 | 0.688 | 0.565 | 0.465 | 0.394 | 0.343 | 0.308 | 0.287 |
| No_Transfer | 0.442 | 0.371 | 0.313 | 0.267 | 0.231 | 0.202 | 0.181 | 0.166 | 0.154 | 0.142 | 0.131 |
| MMSE_CE [29] | 0.589 | 0.499 | 0.416 | 0.337 | 0.272 | 0.226 | 0.193 | 0.172 | 0.157 | 0.146 | 0.141 |
| MMSE_CE [12] | 0.593 | 0.491 | 0.399 | 0.317 | 0.249 | 0.198 | 0.161 | 0.137 | 0.122 | 0.112 | 0.106 |
| **Proposed** | **0.1586** | **0.1067** | **0.0709** | **0.0502** | **0.0359** | **0.0251** | **0.0172** | **0.0117** | **0.0081** | **0.0059** | **0.0049** |

plotted in Fig 4. Two conventional equalization methods, i.e., ZF equalization and MMSE equalization, are utilized to equalize wireless channel. In Fig 4, we compare the BERs among those of [12, 29], and the proposed scheme. With the same CE methods and equalization methods, the BER of [12] is lower than that of [29] due to the influence of CE. Meanwhile, both the "Proposed+ZF_SD" and "Proposed+MMSE_SD" achieve the smallest BER by using the same equalization method for all given SNRs. For the case where SNR = 20dB, the BER of "Proposed+MMSE_SD" is less than $1.8 \times 10^{-3}$ while the BER of "MMSE_CE+MMSE_SD [12]" is about $1.3 \times 10^{-2}$. This verifies that the proposed CE scheme improves the BER performance as well.

As a whole, compared with the "LS_CE [12]", "LS_CE [29]", "No_Transfer", "MMSE_CE [29]", and "MMSE_CE [12]", both the performances of CE and SD in the new environments are improved by "Proposed". Especially, compared with the "MMSE_CE [12]", the "Proposed" can achieve the lower NMSE without the second-order statistics about the channel and noise. Meanwhile, based on the "Proposed", the LS/MMSE equalization obtains the smallest BER due to the improvement of CE.

## Analysis of parameter impact

In this subsection, the robustness of the proposed scheme against parameter variation is analysed. The impact of pilot number $P$ is first discussed, followed by the superposition factor $\rho$. It is worth noting that, besides the change of the impact parameters (i.e, $P$ and $\rho$), other basic parameters remain the same as those given in *CE and SD Performance* during the simulations.

**Impact of $P$.** The NMSE of CE and the BER of SD are usually impacted by the number of pilot (i.e., $P$). To reveal the robustness of the proposed CE scheme against the impact of $P$, the NMSE of CE and the BER of SD are given in Figs 5 and 6, respectively.

Since the "LS_CE [12]" and "MMSE_CE [12]" achieve smaller NMSEs than those of [29] with the same CE method, we only employ [12] as a comparison when discussing the parameter $P$. In Fig 5, $P = 4$, $P = 8$, and $P = 16$ are considered. From Fig 5, the NMSEs of "Proposed", "LS_CE [12]", and "MMSE_CE [12]" decline with the enlargement of the pilot number $P$. That is, the more accurate CSI can be achieved by the conventional CE schemes. In addition, for each given value of $P$, the performance of "Proposed" achieves the smallest NMSE for all given SNRs. From Fig 5, when SNR = 20dB and $P = 4$, the NMSE of "MMSE_CE [12]" is higher than

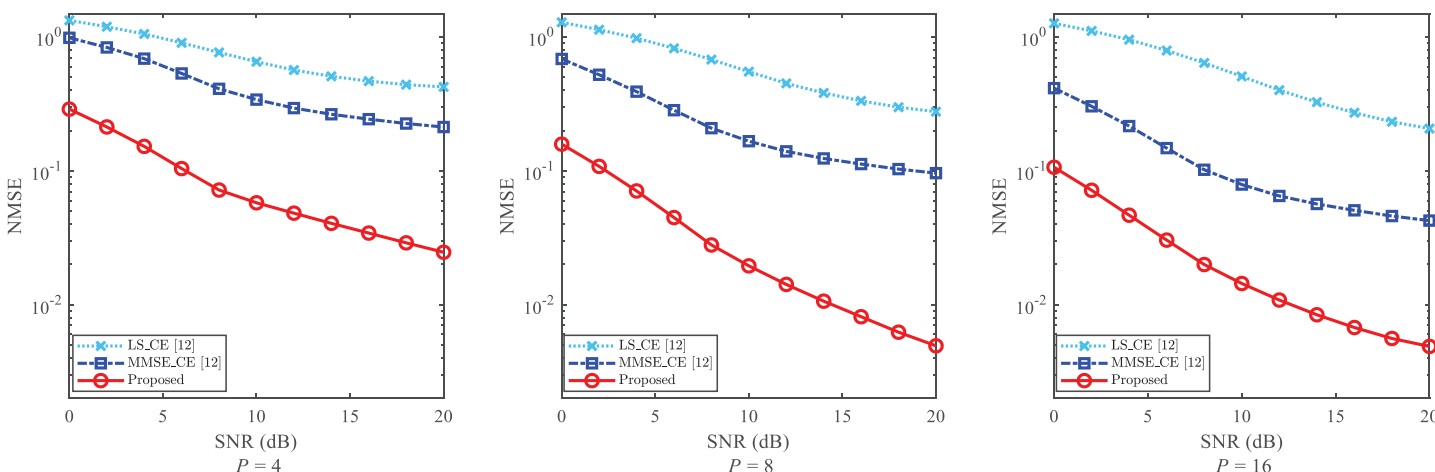

**Fig 5. NMSE of CE against the impact of $P$, where $P = 4$, $P = 8$ and $P = 16$ are considered, respectively.**

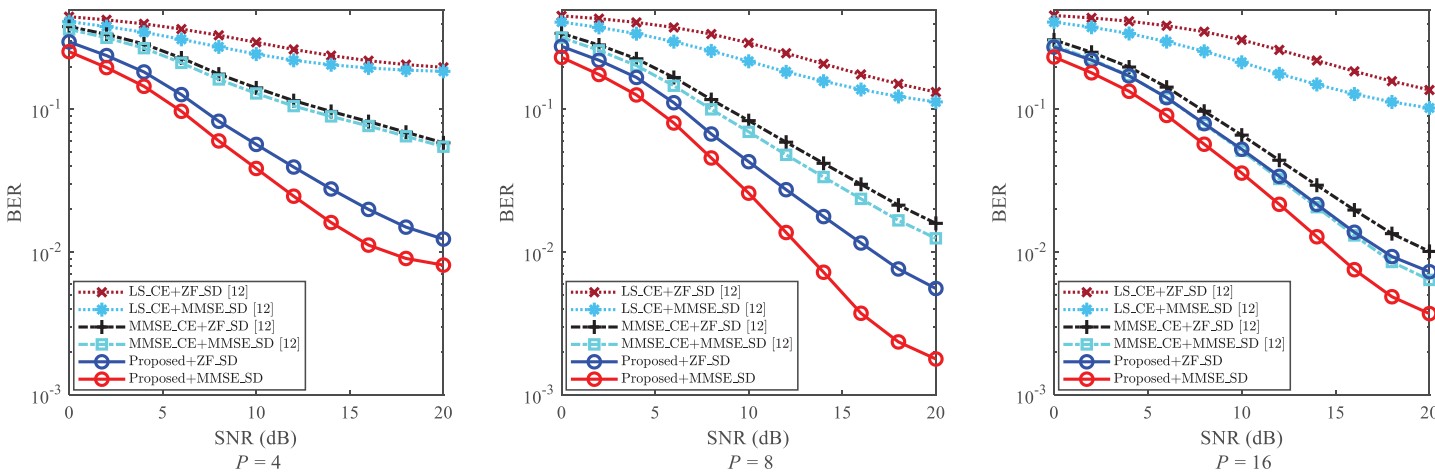

**Fig 6. BER of SD against the impact of *P*, where *P* = 4, *P* = 8 and *P* = 16 are considered, respectively.**

$2 \times 10^{-1}$, while the NMSE of "Proposed" is lower than $3 \times 10^{-2}$. This reflects that the proposed scheme improves the NMSE compared with the existing methods with the variations of *P*. Meanwhile, it could be observed that proposed scheme has its robustness against the varying *P*.

Fig 6 gives the BER performance with the different equalization methods against the impact of *P*. From Fig 6, the varying of BER is not regular. The reason is that the performance of SD is affected by both *P* and the performance of CE. Thus, to achieve a lower BER, the pilot number should be trade off in the proposed scheme. Nevertheless, for each given *P*, the BERs of "Proposed+ZF_SD" and "Proposed+MMSE_SD" obtain smaller BERs than those of [12] with the same SD methods. For example, for the cases where SNR = 20dB and *P* = 8, the BER of "Proposed + MMSE_SD" is about $1.6 \times 10^{-3}$, while the BER of "MMSE_CE+MMSE_SD [12]" is larger than $1 \times 10^{-2}$. This validates that the proposed scheme improves the SD performance and has its robustness against the varying of *P*.

On a whole, from Figs 5 and 6, the proposed scheme reaches lower NMSEs and BERs than those of [12]. With the varying *P*, the proposed scheme still improves the performance of CE and SD, and thus possesses its robustness.

**Impact of *ρ*.** Usually, the CE's NMSE is influenced by the power proportional coefficient, i.e., *ρ*. To validate the robustness against the impact of *ρ*, the NMSE of CE is illustrated in Fig 7 with different values of *ρ* (i.e., *ρ* = 0.1, *ρ* = 0.2, and *ρ* = 0.3).

Similar to the reason of the simulation against parameter *P*, we only compare the performance of [12]. From Fig 7, the CE's NMSEs of "Proposed", "LS_CE [12]", and "MMSE_CE [12]" decrease with the enlargement of the power proportional coefficient *ρ*. Although the decline of NMSE is not obvious when compared *ρ* = 0.3 with *ρ* = 0.2, the tendency of decreasing is still observed. The reason for this phenomenon is that the performance of CE is improved due to the increased pilot power. Moreover, the larger the pilot power employed, the higher accuracy the CE obtained. Besides, for the cases where *ρ* = 0.1, *ρ* = 0.2, and *ρ* = 0.3, the "Proposed" obtains a smaller NMSE than those of "LS_CE [12]" and "MMSE_CE [12]". For example, for the cases where SNR = 20dB and *ρ* = 0.1, the NMSE of "MMSE_CE [12]" is higher than $1 \times 10^{-1}$, while the NMSE of "Proposed" is lower than $1 \times 10^{-2}$. This reflects that the "Proposed" reduces the NMSE of CE against the varying *ρ*, and thus possesses its robustness against the impact of *ρ*.

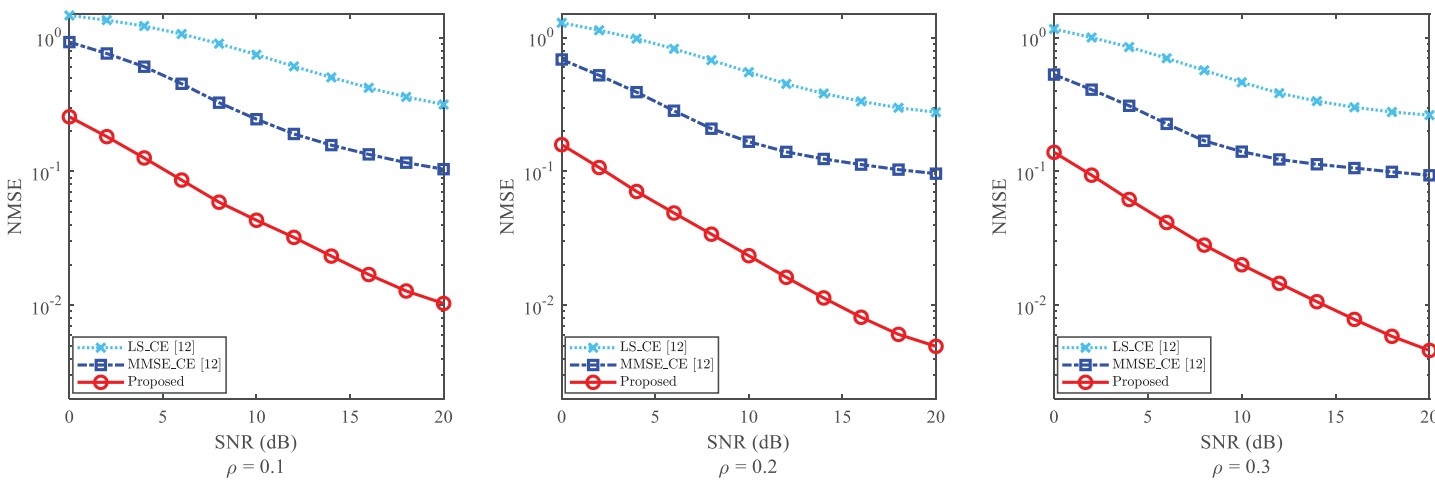

**Fig 7. NMSE of CE against the impact of $\rho$, where $\rho = 0.1$, $\rho = 0.2$ and $\rho = 0.3$ are considered, respectively.**

In Fig 8, the BERs of SD are obtained based on three CE methods, i.e., "Proposed", "LS_CE [12]", and "MMSE_CE [12]". From Fig 8, with the increase of $\rho$, the BERs of the different methods do not change greatly. Thus, although the power factor $\rho$ influences the performance of CE, it has a little impact on SD due to the superimposed pilots on the data-nulling in DNSP scheme. Even so, the significant improvement of BER performance is still obtained. For each given $\rho$, the BERs of "Proposed+ZF_SD" and "Proposed+MMSE_SD" obtain smaller BERs than those of SD methods in [12]. From Fig 8, when SNR = 20dB and $\rho = 0.2$, the BER of "Proposed+MMSE_SD" is lower than $2 \times 10^{-3}$, while the BER of "MMSE_CE+MMSE_SD [12]" is about $1.2 \times 10^{-2}$. This indicates that the proposed scheme improves the SD performance even against the varying of $\rho$.

To sum up, the NMSEs and BERs of the proposed scheme show superiority over those of [12] in Figs 7 and 8, respectively. Against the impact of $\rho$, the proposed scheme can effectively reduce the NMSE of CE and the BER of SD to possess its robustness.

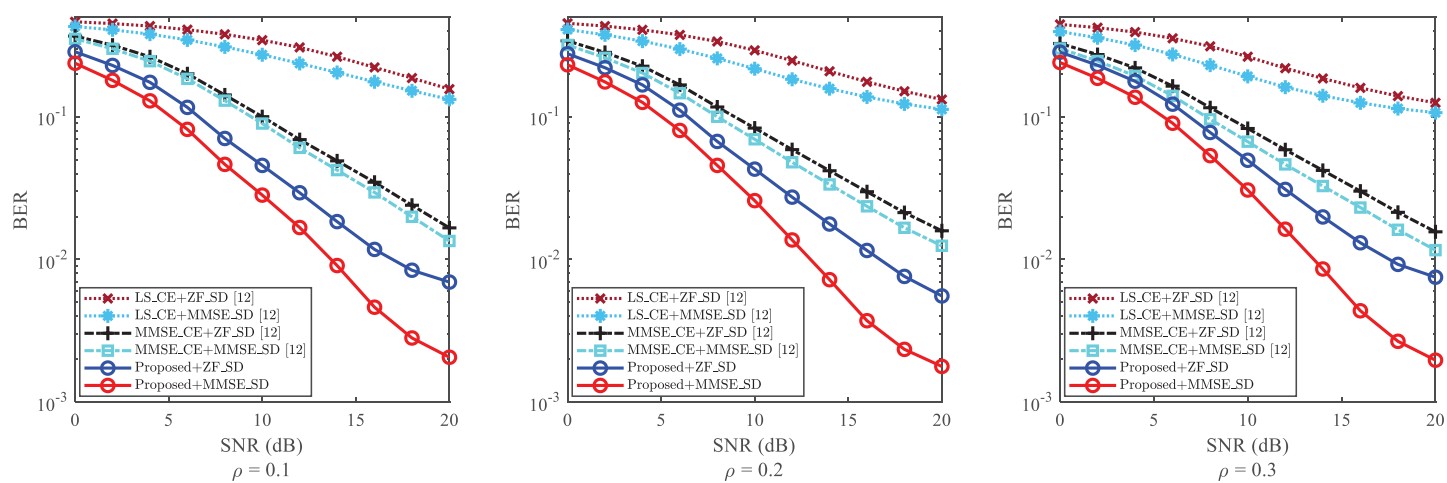

**Fig 8. BER of SD against the impact of $\rho$, where $\rho = 0.1$, $\rho = 0.2$ and $\rho = 0.3$ are considered, respectively.**

## Conclusion

In this paper, the TL-based CE in OFDM systems by using DNSP scheme has been investigated and thus forms a novel network ReCNN. In the proposed scheme, the employed CNN improves the accuracy of DL-based CE by fusing the linear and nonlinear solutions. With the lightweight TL network, the CE generalization is enhanced. To this end, not only the improvement of CE accuracy is achieved by DL-based mode, but also the issue of model mismatch is addressed by TL approach without significant increase of computational complexity. Compared with existing DNSP schemes with MMSE-based CE, the proposed scheme obtains the lower NMSE and BER without the requirements of second-order statistics about the channel and noise. With environmental changes, the CE generalization is also validated. The proposed scheme presents a good estimation accuracy and model generalization, promoting the existing researches of DL-based CE move towards practical application. In future works, we will investigate the online learning-based CE for the DNSP scheme in OFDM systems.

## Supporting information

**S1 Fig.**
(PDF)

**S2 Fig.**
(PDF)

**S3 Fig.**
(PDF)

**S4 Fig.**
(PDF)

**S5 Fig.**
(PDF)

**S6 Fig.**
(PDF)

**S7 Fig.**
(PDF)

**S8 Fig.**
(PDF)

**S1 File.**
(RAR)

## Author Contributions

**Conceptualization:** Chaojin Qing.

**Data curation:** Chaojin Qing, Lei Dong.

**Formal analysis:** Chaojin Qing, Li Wang, Guowei Ling, Jiafan Wang.

**Funding acquisition:** Chaojin Qing.

**Investigation:** Chaojin Qing.

**Methodology:** Chaojin Qing.

**Project administration:** Chaojin Qing.

**Resources:** Chaojin Qing, Lei Dong.

**Software:** Lei Dong.

**Supervision:** Chaojin Qing, Lei Dong, Li Wang, Guowei Ling, Jiafan Wang.

**Validation:** Chaojin Qing, Lei Dong, Li Wang, Guowei Ling, Jiafan Wang.

**Writing – original draft:** Lei Dong.

**Writing – review & editing:** Chaojin Qing, Lei Dong, Li Wang, Guowei Ling, Jiafan Wang.

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
