## [Decision Letter · Decision Letter 0]

14 Dec 2021

PONE-D-21-33229Deep Transfer Learning-based Channel Estimation in OFDM Systems Using Data-nulling Superimposed PilotsPLOS ONE

Dear Dr. Qing,

Thank you for submitting your manuscript to PLOS ONE. After careful consideration, we feel that it has merit but does not fully meet PLOS ONE’s publication criteria as it currently stands. Therefore, we invite you to submit a revised version of the manuscript that addresses the points raised during the review process.

We look forward to receiving your revised manuscript.

Kind regards,

Nguyen Quoc Khanh Le

Academic Editor

PLOS ONE

Journal Requirements:

2.Please note that PLOS ONE has specific guidelines on code sharing for submissions in which author-generated code underpins the findings in the manuscript. In these cases, all author-generated code must be made available without restrictions upon publication of the work. Please review our guidelines at https://journals.plos.org/plosone/s/materials-and-software-sharing#loc-sharing-code and ensure that your code is shared in a way that follows best practice and facilitates reproducibility and reuse.

3. Thank you for submitting the above manuscript to PLOS ONE. During our internal evaluation of the manuscript, we found significant text overlap between your submission and the following previously published works, some of which you are an author.

-https://ieeexplore.ieee.org/document/9395511/

Please revise the manuscript to rephrase the duplicated text, cite your sources, and provide details as to how the current manuscript advances on previous work. Please note that further consideration is dependent on the submission of a manuscript that addresses these concerns about the overlap in text with published work.

"This work is supported in part by the Sichuan Science and Technology Program (Grant No. 2021JDRC0003), the Major Special Funds of Science and Technology of Sichuan Science and Technology Plan Project (Grant No. 19ZDZX0016 /2019YFG0395), the Demonstration Project of Chengdu Major Science and Technology Application (Grant No. 2020-YF09- 00048-SN), the Key Scientific Research Fund of Xihua University (Grant No. Z1120941), and the Special Funds of Industry Development of Sichuan Province (Grant No. zyf-2018-056)."

Reviewers' comments:

Reviewer's Responses to Questions

**Comments to the Author**

1. Is the manuscript technically sound, and do the data support the conclusions?

Reviewer #1: Yes

Reviewer #2: Yes

Reviewer #3: Yes

Reviewer #4: No

2. Has the statistical analysis been performed appropriately and rigorously? 

Reviewer #1: No

Reviewer #2: Yes

Reviewer #3: Yes

Reviewer #4: No

3. Have the authors made all data underlying the findings in their manuscript fully available?

Reviewer #1: Yes

Reviewer #2: Yes

Reviewer #3: Yes

Reviewer #4: No

4. Is the manuscript presented in an intelligible fashion and written in standard English?

Reviewer #1: Yes

Reviewer #2: Yes

Reviewer #3: Yes

Reviewer #4: No

5. Review Comments to the Author

Reviewer #1: The Research Paper needs the following Major revisions and is subject for re-review, and after re-review, the final decision for the paper will be taken:

1. Make the abstract better and clearly state what is the aim/objective of the paper, and what novelty is connected and what experimental results are observed in what %age and in what parameters.

2. Add more information to the Introduction section with regard to Problem Definition and Scope of the paper, and even add some more technical highlights covering the related information to the main theme of the paper.

3. Add Contributions at the end of Introduction. Add Organization of the paper.

4. Under related works, add min 15-25 papers and every paper should be highlighted with what is being proposed, what is the novelty and what experimental results are observed. At the end, highlight in 9-15 lines what overall technical gaps are observed that led to the design of the proposed methodology.

5. Add flowchart of the proposed methodology.

6. Under experimentation, Add the information of the Dataset.

7. Stress more on the Results and add Tables with Data values on the basis of which graphs are plotted.

8. Add performance comparison of the proposed work with existing techniques.

9. Add case study based discussion to the paper.

10. Add future scope to this paper.

Reviewer #2: The abstract should be revised as it does not enough chiefly introduce the area of research along with the research question.

Introduction should be rewritten in a professional way.

Please explain the proposed method in more details, what is the novelty of the proposed method compared to the state of the art?

Current experiments are weak and fair comparison with other recent methods is very necessary.

I think it would be good to represent the experiment results in the abstract.

The experiment description section is too rough. The description of data collection should be added.

For the experimental results, it will be good to present a statistical test in the comparison of the results with other published methods.

This can help to support the claim on improved results obtained with the selection methods studied.

Other aspect where paper can be improved is motivation and the reason for the given architecture.

The current approach seems to be more like we have this different types of architectures, let's mix them and present results by training them.

It would be of great interest why a particular model was selected and what a particular part in the framework is helping us to learn.

English can be improved. Proofreading should ensure appropriate use of grammar, tense and punctuations.

Longer sentences should be converted into smaller ones.

Reviewer #3: This paper investigates deep transfer NN to solve the issues of channel estimation for DNSP systems. The paper is generally well written, logical and discusses a novel topic. Still, there are some defects which are need to be revised. Please see my concerns below.

1. In the abstract, the authors claim that DNSP has high spectrum efficiency. Please elaborate on why?

2. How to design the unitary matrix $mathbf{W}$ in Equation (2). What are the specific requirements?

3. In the training phase, is the verification set generated and used to search the optimized parameters? If the validation set is adopted, the corresponding processing should be described.

4. The author mentioned that after pre-training, if you change the test environment, you also need to fine-tune the network. So, what is the significance of such transfer learning?

5.In the first sentence of the Definition 1 on page 4, the author gave a marginal probability distribution $P\\left( h \\left| h_{\\mathrm{LS}} \\right) \\right.$, but the $h$ was not explained.

6. In the first sentence under Equation (1), the rounding symbol $\\lfloor \\cdot \\rfloor$ was not explained in the Notations.

Reviewer #4: 1. I suggest removing the acronym from the title of the manuscript. Similarly, the abstract is packed with multiple acronyms which make the read challenging. To this end, the authors should move all the acronyms from the abstract to the main body of the manuscript, and should make sure that every acronym is defined exactly once at its first use.

2. The description of the generated datasets, and the division into training/test samples is vague – the authors should make sure that the entire experimentation is fully reproducible and possible to understand based solely on the description provided in the manuscript. Additionally, the authors should make their data publicly available so that other research groups can reproduce the experiments (the same applies to the implementation of their deep learning model).

3. The authors should confront their technique with other approaches from the literature (also, I suggest performing multi-fold cross-validation experiments to understand the abilities of the introduced techniques). Finally, the authors should back up their claims with appropriate statistical testing.

4. The authors should perform careful proofreading of the entire manuscript – there are quite a number of grammar issues around. Additionally, please avoid having the paragraphs that contain just one sentence. Finally, please avoid using short forms (“doesn’t”).

6. PLOS authors have the option to publish the peer review history of their article (what does this mean?). If published, this will include your full peer review and any attached files.

Reviewer #1: No

Reviewer #2: **Yes: **Dr B Santhosh Kumar

Reviewer #3: **Yes: **Guan Gui

Reviewer #4: No

---

## [Author Response · Author response to Decision Letter 0]

24 Jan 2022

We have studied the reviewers’ comments carefully and made revisions and upload this response as a separate file labeled 'Response to Reviewers'.

---

## [Decision Letter · Decision Letter 1]

4 Apr 2022

PONE-D-21-33229R1Transfer Learning-based Channel Estimation in Orthogonal Frequency Division Multiplexing Systems Using Data-nulling Superimposed PilotsPLOS ONE

Dear Dr. Qing,

Thank you for submitting your manuscript to PLOS ONE. After careful consideration, we feel that it has merit but does not fully meet PLOS ONE’s publication criteria as it currently stands. Therefore, we invite you to submit a revised version of the manuscript that addresses the points raised during the review process.

We look forward to receiving your revised manuscript.

Kind regards,

Nguyen Quoc Khanh Le

Academic Editor

PLOS ONE

Journal Requirements:

Reviewers' comments:

Reviewer's Responses to Questions

**Comments to the Author**

1. If the authors have adequately addressed your comments raised in a previous round of review and you feel that this manuscript is now acceptable for publication, you may indicate that here to bypass the “Comments to the Author” section, enter your conflict of interest statement in the “Confidential to Editor” section, and submit your "Accept" recommendation.

Reviewer #1: All comments have been addressed

Reviewer #2: All comments have been addressed

Reviewer #4: All comments have been addressed

2. Is the manuscript technically sound, and do the data support the conclusions?

Reviewer #1: Yes

Reviewer #2: Yes

Reviewer #4: (No Response)

3. Has the statistical analysis been performed appropriately and rigorously? 

Reviewer #1: Yes

Reviewer #2: Yes

Reviewer #4: (No Response)

4. Have the authors made all data underlying the findings in their manuscript fully available?

Reviewer #1: Yes

Reviewer #2: Yes

Reviewer #4: (No Response)

5. Is the manuscript presented in an intelligible fashion and written in standard English?

Reviewer #1: Yes

Reviewer #2: Yes

Reviewer #4: (No Response)

6. Review Comments to the Author

Reviewer #1: The Revised paper has incorporated all the revisions in proper manner. And now the paper stands Accepted with no further revisions.

Reviewer #2: Experimental results show very good accuracy and performance Also, proposed methodology represents an improved efficiency which can be useful for society in future

Reviewer #4: I am happy to see that the authors have indeed seriously addressed my concerns, and the manuscript is in better shape now. I would, however, encourage the authors to do yet another pass through the entire paper---please avoid using vague statements in the discussion part ("performance is slightly worse"), as they should be objective. Also, it would be great to avoid having the paragraphs that contain just one sentence.

7. PLOS authors have the option to publish the peer review history of their article (what does this mean?). If published, this will include your full peer review and any attached files.

Reviewer #1: No

Reviewer #2: **Yes: **Dr B Santhosh Kumar

Reviewer #4: No

---

## [Author Response · Author response to Decision Letter 1]

21 Apr 2022

We appreciate reviewers very much for your positive and constructive comments. We have studied the reviewers’ comments carefully and made revisions which are marked in yellow in the paper.

---

## [Editor Report · Decision Letter 2]

12 May 2022

Transfer Learning-based Channel Estimation in Orthogonal Frequency Division Multiplexing Systems Using Data-nulling Superimposed Pilots

PONE-D-21-33229R2

Dear Dr. Qing,

We’re pleased to inform you that your manuscript has been judged scientifically suitable for publication and will be formally accepted for publication once it meets all outstanding technical requirements.

Kind regards,

Nguyen Quoc Khanh Le

Academic Editor

PLOS ONE